# Associations of Plastic Bottle Exposure with Infant Growth, Fecal Microbiota, and Short-Chain Fatty Acids

**DOI:** 10.3390/microorganisms11122924

**Published:** 2023-12-05

**Authors:** Curtis Tilves, Heather Jianbo Zhao, Moira K. Differding, Mingyu Zhang, Tiange Liu, Cathrine Hoyo, Truls Østbye, Sara E. Benjamin-Neelon, Noel T. Mueller

**Affiliations:** 1Department of Epidemiology, Johns Hopkins Bloomberg School of Public Health, Baltimore, MD 21205, USA; curtis.tilves@cuanschutz.edu (C.T.); heatherzhao1@gmail.com (H.J.Z.); mdiffer1@jhu.edu (M.K.D.); mingyu_zhang@harvardpilgrim.org (M.Z.); tliu57@jhu.edu (T.L.); 2Lifecourse Epidemiology of Adiposity and Diabetes (LEAD) Center, University of Colorado Anschutz Medical Campus, Aurora, CO 80045, USA; 3Temerty Faculty of Medicine, University of Toronto, Toronto, ON M5T 3M7, Canada; 4Department of Population Medicine, Harvard Pilgrim Health Care Institute, Harvard Medical School, Boston, MA 02215, USA; 5Department of Biological Sciences, North Carolina State University, Raleigh, NC 27695, USA; choyo@ncsu.edu; 6Department of Family Medicine and Community Health, Duke University, Durham, NC 27708, USA; truls.ostbye@duke.edu; 7Department of Health, Behavior and Society, Johns Hopkins Bloomberg School of Public Health, Baltimore, MD 21205, USA; sara.neelon@jhu.edu; 8Department of Pediatrics Section of Nutrition, University of Colorado Anschutz Medical Campus, Aurora, CO 80045, USA

**Keywords:** anthropometry, plastic, microbiome, short-chain fatty acids

## Abstract

Background/Objectives: Murine models show that plastics, via their chemical constituents (e.g., phthalates), influence microbiota, metabolism, and growth. However, research on plastics in humans is lacking. Here, we examine how the frequency of plastic bottle exposure is associated with fecal microbiota, short-chain fatty acids (SCFAs), and anthropometry in the first year of life. Subjects/Methods: In 442 infants from the prospective Nurture birth cohort, we examined the association of frequency of plastic bottle feeding at 3 months with anthropometric outcomes (skinfolds, length-for-age, and weight-for-length) at 12 months of age and growth trajectories between 3 and 12 months. Furthermore, in a subset of infants (*n* = 70) that contributed fecal samples at 3 months and 12 months of age, we examined plastic bottle frequency in relation to fecal microbiota composition and diversity (measured by 16S rRNA gene sequencing of V4 region), and fecal SCFA concentrations (quantified using gas chromatography mass spectrometry). Results: At 3 months, 67.6% of infants were plastic bottle fed at every feeding, 15.4% were exclusively breast milk fed, and 48.9% were exclusively formula fed. After adjustment for potential confounders, infants who were plastic bottle fed less than every feeding compared to those who were plastic bottle fed at every feeding at 3 months did not show differences in anthropometry over the first 12 months of life, save for lower length-for-age z-score at 12 months (adjusted β = −0.45, 95% CI: −0.76, −0.13). Infants who were plastic bottle fed less than every feeding versus every feeding had lower fecal microbiota alpha diversity at 3 months (mean difference for Shannon index: −0.59, 95% CI: −0.99, −0.20) and lower isovaleric acid concentration at 3 months (mean difference: −2.12 μmol/g, 95% CI: −3.64, −0.60), but these results were attenuated following adjustment for infant diet. Plastic bottle frequency was not strongly associated with microbiota diversity or SCFAs at 12 months after multivariable adjustment. Frequency of plastic bottle use was associated with differential abundance of some bacterial taxa, however, significance was not consistent between statistical approaches. Conclusions: Plastic bottle frequency at 3 months was not strongly associated with measures of adiposity or growth (save for length-for-age) over the first year of life, and while plastic bottle use was associated with some features of fecal microbiota and SCFAs in the first year, these findings were attenuated in multivariable models with infant diet. Future research is needed to assess health effects of exposure to other plastic-based products and objective measures of microplastics and plastic constituents like phthalates.

## 1. Introduction

Exposure to plastics, especially during vulnerable periods of development (e.g., infancy), is a growing public health concern [1]. A recent study estimated that infants have significantly higher fecal concentrations of plastic materials (daily intake of 83,000 ng/kg body weight) compared to adults (5800 ng/kg body weight) [2], which is thought to be related to plastic-based products during infant feeding, such as bottle feeding and sippy cups [2,3]. Although childhood exposure to plasticizers (i.e., phthalates) has been associated with higher childhood body mass index (BMI) and waist circumference [4,5,6], to our knowledge, the association of infant plastic-product exposure and growth and adiposity in the first year of life has not been reported. Furthermore, there are limited data on potential mechanisms by which plastic exposure might alter health outcomes.

Human gut microbiota, the ensemble of microorganisms living in the intestinal tract, are largely determined by environmental factors [7]. Studying gut microbiota provides an opportunity to examine whether the health effects of environmental exposures, like exposure to plastic, start in the intestine. Human studies suggest that infant fecal microbiota features, such as diversity and composition, are associated with altered metabolism and growth [8,9,10,11,12]. Gut microbiota is purported to affect metabolic outcomes by producing metabolites, such as short-chain fatty acids (SCFAs) [13]. To our knowledge, no studies have examined associations of plastic exposure in infancy with gut microbiota or SCFAs.

The primary aim of this study was to examine associations of plastic bottle exposure frequency with measures of growth and adiposity in the first year of life. Secondary aims of this study were to examine associations of plastic bottle exposure frequency with infant fecal microbiota and fecal SCFAs. We hypothesized that plastic bottle frequency is associated with adiposity measures in the first year as well as differences in fecal microbiota diversity, composition, and SCFAs.

## 2. Methods

### 2.1. Study Population

We used data from the prospective Nurture birth cohort from central North Carolina [14], which recruited participants from 2013 to 2015. The Nurture birth cohort enrolled 666 women with a singleton pregnancy at 20–36 weeks of gestation from a county health department or private prenatal clinic. Mothers were included if they were at least 18 years of age, had a singleton pregnancy, would live in the area a minimum of one year after delivery, and were able to speak and read English. Mother–infant pairs were excluded if the infant was delivered prior to 28 weeks of gestation, had congenital abnormalities, or required 3 or more weeks of hospitalization postnatally. We received maternal written informed consent at recruitment and reconfirmed via phone shortly after delivery. The Nurture birth cohort study followed the guidelines of the Declaration of Helsinki and procedures involving human subjects and was approved by the Duke University Medical Center Institutional Review Board (human subjects committee) (PRO0036342). The study is registered at clinicaltrials.gov (accessed on 29 November 2023) (NCT01788644).

### 2.2. Primary Exposure—Plastic Bottle Frequency

We asked mothers to report the daily frequency of plastic bottle feeding for their child every month up until 12 months during automated interactive voice response calls. Our analyses are based on plastic bottle feeding at 3 months due to limited reporting beyond this time. We grouped infant’s plastic bottle use into 6 categories: <1 time/day, 1 time/day, 2–3 times/day, 4–5 times/day, 5+ times/day but not at every feeding, or at every feeding. In the analysis of the growth trajectories, we classified plastic bottle frequency into four groups: <1 time/day, 1–3 times/day, 4+ times/day, and every feeding. In analyses for microbiota and SCFAs, because of limited sample size, we classified plastic bottle frequency into less than every feeding vs. every feeding.

### 2.3. Infant Anthropometry

We measured infant weight, height, and skinfold thicknesses at home visits conducted at 3, 6, 9, and 12 months of age. Trained data collectors measured infant weight in light clothing without shoes using a Seca Infant Scale (to nearest 1/8 pound) and infant height via a ShorrBoard Portable Length Board (to the nearest 0.1 inch). We rounded infant abdomen, subscapular, and triceps skinfold thicknesses measurements to the nearest millimeter using standard techniques [15]. We repeated all measurements three times to reduce measurement error with the final measurement recorded as the average of the three values. We calculated the age- and sex-specific weight-for-length z-score, the BMI-for-age z-score, and length-for-age z-score using the World Health Organization Child Growth Standards [16]. We summed subscapular and triceps skinfolds thickness measures as a proxy for overall adiposity.

### 2.4. Stool Sample Collection, DNA Extraction, Amplification, and 16S rRNA Gene Sequencing

We collected stool at 3- and 12-month home visits from diapers in a subset of infants (*n* = 70). We transferred stool samples to 2 mL cryogenic vials (ThermoFisher, Waltham, MA, USA) and froze the vials at −80 °C until they were ready for processing. We sent frozen fecal samples on dry ice to Microbiome Insights (Vancouver, BC, Canada) for 16S rRNA sequencing using Illumina MiSeq. Their laboratory technicians thawed specimens and extracted DNA using the QIAgen MagAttract PowerSoil for KingFisher (Qiagen, Venlo, The Netherlands). They placed 0.5 g of stool in each bead plate well of the PowerSoil kit and extracted DNA per the manufacturer’s instructions. Then, they quantified DNA with the Quant-iT dsDNA high-sensitivity kit (ThermoFisher, Waltham, MA, USA). They used Thermo Phusion Hot Start II DNA Polymerase (ThermoFisher Cat. No. F549S) to conduct polymerase chain reaction (PCR) amplification. The extracted 1–10 ng of DNA were started in the PCR at 98 °C for 2 min with subsequent 30 cycles at 98 °C for 20 s, 55 °C for 15 s, 72 °C for 30 s, and at 72 °C for 10 min. The technicians prepared libraries following procedures described by Kozich et al. [17]. They normalized PCR product using the SepalPrep Normalization Prep Plate Kit (TheromFisher Cat. No. A1051001), pooled 5 µL of each normalized sample into a single library, and concentrated pools using the DNA Clean and Concentrator kit (Zymo Cat. No. D4013). They then purified amplicon pools using the Qiagen QIAquick Gel Extraction Kit (Qiagen Cat. No. 28706), diluted them to 4 nM, and denatured them with 2.0 N NaOH. The technicians then sequenced the V4 region of the 16S rRNA gene using the Illumina MiSeq platform (Illumina, San Diego, CA, USA) with a MiSeq 500 Cycle V2 Reagent Kit (Illumina Cat. No. MS-102-2003). They used primers and sample-specific barcode demultiplexing as described in Kozich et al. [17].

### 2.5. 16S rRNA Gene Sequence Pre-Processing and Taxonomic Assignment

We demultiplexed sequences and used the R package DADA2 (v1.8), following author recommendations, to perform quality control and to resolve amplicon sequence variants (ASVs) [18]. Extended details of sequence pre-processing steps, including quality scores for forward and reverse reads, trimming, error model estimation, dereplication, denoising, and chimera removal can be found in our prior publications from this cohort [19,20]. We assigned taxonomy using the human intestinal taxa from the HITdb v.1.00 16S rRNA sequence database and the dada2 function “assignTaxonomy” [21].

### 2.6. Phylogenetic Tree Generation

We used the de-noised ASVs to generate a phylogenetic tree following the recommendations of a Bioconductor workflow [22]. Briefly, we aligned ASVs and created a neighbor-joining tree using the R package DECIPHER [23]. We created a generalized time-reversible (with gamma rate variation) maximum likelihood phylogenetic tree using the R package phangorn [24]; we rooted this tree at the midpoint. We then merged all taxonomic, phylogenetic tree, and sample metadata using the R package phyloseq [25].

### 2.7. Measurement of SCFA Metabolites

Technicians at Microbiome Insights (Vancouver, BC, Canada) measured fecal SCFA concentrations using gas chromatography (Thermo Trace 1310) paired to a flame ionization detector (Thermo) following previously published methods [26]. Briefly, they re-thawed stool samples (following 16s rRNA sequencing), resuspended them in MilliQ-grade water at 4.0 m/s for 1 min, and homogenized them using an MP Bio FastPrep. They acidified samples to a pH of 2.0 using additions of 5 M hydrochloric acid, incubated them for 10 min, and centrifuged at them 10,000 revolutions per minute (~9633 g, Sorvall Legend Micro 21R). They extracted the supernatant and spiked it with 2-Ethylbutyric acid to a concentration of 1 mM and directly injected it into a Thermo TG-WAXMS A GC Column (30 m, 0.32 mm, 0.25 μm). They used a standard solution of SCFAs to obtain individual calibration curves. They quantified the SCFAs acetic acid, propionic acid, butyric acid, isobutyric acid, valeric acid, isovaleric acid, and hexanoic acid; we summed these individual SCFAs to generate the total amount of SCFAs.

### 2.8. Other Data Measurements

We abstracted data on delivery mode (Caesarean section vs. vaginal delivery), birth weight (kg), infant sex, and gestational age at birth (weeks) from medical records. We asked mothers to self-report their age (year), pre-pregnancy weight (kg) and height (m), highest education obtained (categorized as high school or below vs. post-secondary), and their infant’s race and ethnicity. We asked mothers to self-report the number of persons living in their household, their household income (categorized as <$20,000/year vs. ≥$20,000/year), and current smoking status (yes vs. no) at each home visit. We used self-reported data from the 3-month visit in our analyses; for mothers missing household income at 3 months, we substituted with their income reported at birth.

We asked mothers to self-report infant feeding practices at home visits and during monthly automated interactive voice response calls between home visits. At these assessments, we asked mothers about the feeding method used in the past month (exclusive formula feeding, exclusive breast milk feeding, or mixed formula and breast milk feeding). We calculated the duration of exclusive breast milk feeding as the total number of months in which mothers reported exclusive breast milk feeding up until month 3 (range: 0–3 months).

### 2.9. Data Analysis

#### 2.9.1. Analytic Sample Sizes and Model Construction

To maximize analytic power, we performed analyses using the largest available sample sizes. A study flow diagram is provided to indicate sample sizes for each analysis (Appendix A). For all analyses, we restricted the analytic sample to participants who had non-missing data for 3-month plastic bottle exposure, non-missing key covariate data (birth weight (kg), gestational age (weeks), maternal age (years), household income (<$20,000/year vs. ≥$20,000/year)), and non-missing infant dietary data (duration of exclusive breastfeeding (months) and feeding method at 3 months (exclusive breast milk, exclusive formula, or mixed feeding)). This resulted in a maximum of *n* = 442 participants for anthropometric growth models, *n* = 66 for SCFA models, and *n* = 63 for microbiome models. Models were further restricted to the most complete data for each outcome and timepoint (described in the sections below).

In general, for all analyses, we considered associations of plastic bottle frequency with an outcome of interest using the following model constructions: Model 1 (unadjusted); Model 2 (adjusted for key covariates): Model 1 + birth weight, gestational age, maternal age, and household income; Model 3 (further adjusted for infant diet): Model 2 + duration of exclusive breastfeeding, feeding method at 3 months. Alterations to this modeling schema (e.g., inclusion of interaction terms) are described in the sections below.

#### 2.9.2. Anthropometry Analyses

We report participant characteristics, both overall and by plastic bottle frequency at 3 months, for our largest analytic sample (*n* = 442) using mean (SD), median (interquartile range), or N (%) as appropriate. We also present the mean (SD) of each anthropometric outcome by timepoint and plastic bottle frequency at 3 months.

We assessed the association of plastic bottle frequency at 3 months (<1 time/day, 1–3 times/day, 4+ times/day but not every feeding, and at every feeding) with anthropometric outcomes (subscapular skinfolds, triceps skinfolds, abdominal skinfolds, subscapular + triceps skinfolds, length-for-age z-score, BMI-for-age z-score, and weight-for-length z-score) at 12 months of age (*n* = 358–360). We then examined the association of 3-month plastic bottle frequency (<every feeding vs. every feeding) with growth trajectories of the above anthropometric outcomes at 3, 6, 9, and 12 months using linear mixed models (*n* = 442). All linear mixed models were assessed using Model 3. We additionally included a random intercept and a time-by-plastic bottle use interaction. We first fit models using maximum likelihood and determined the appropriateness of a linear or quadratic fit using a likelihood ratio test *p* < 0.05. We re-analyzed the best-fit model using restricted maximum likelihood to estimate fixed effects to generate trajectory plots; we present interaction terms based on linear fit models.

### 2.10. Microbiota Analyses

#### 2.10.1. Diversity Analyses

Prior to microbial diversity analyses, we rarefied samples to a sequencing depth of 9000 reads; we rarefied 10 times without replacement and used the average of these rarefactions (rounded to the nearest integer) to estimate alpha and beta diversity metrics. This procedure removed 19 ASVs from diversity analyses, yielding a total of 764 ASVs.

Alpha diversity metrics reflect the biodiversity within an individual sample and can reflect richness (e.g., number of unique sequence variants), evenness (the distribution of sequence variants within a sample), or richness weighted by evenness. For alpha diversity metrics, we calculated the observed ASVs (a measure of richness) and the Shannon diversity index [27] (richness weighted by evenness) using the R package phyloseq [25] and Pielou’s evenness [28] using the R package microbiome [29]. We conducted separate multiple linear regression analyses to estimate the association of plastic bottle frequency at 3 months (<every feeding vs. every feeding) with alpha diversity metrics as dependent variables at 3 (*n* = 62) and 12 months (*n* = 45). We then examined associations of 3-month plastic bottle frequency (<every feeding vs. every feeding) with each alpha diversity metric across the 3- and 12-month timepoints using linear mixed models and Model 3 adjustments (*n* = 63). In the observed ASVs models, we additionally adjusted for unrarefied sequencing depth. We included terms for a random intercept and a time-by-plastic bottle use interaction, and we estimated the fixed effects using the restricted maximum likelihood approach.

Beta diversity metrics reflect the dissimilarities in pairwise sample compositions. We used the R package phyloseq to estimate Bray–Curtis dissimilarity [30] and the R package rbiom [31] to calculate unweighted UniFrac [32] and weighted UniFrac [33] distances; the latter two incorporate phylogenetic dissimilarity. We assessed the clustering of microbiota samples at 3 months and at 12 months by plastic bottle frequency at 3 months (<every feeding vs. at every feeding) using a principal coordinate analysis plot. We used the permutational multivariate analysis of variance (PERMANOVA) method [34] from the adonis2 function in the R package vegan [35] with 9999 permutations to test for differences in beta diversity metrics at 3 (*n* = 62) and 12 months (*n* = 45).

#### 2.10.2. Differential Abundance Analysis

Prior to differential abundance testing, we removed ASVs with low prevalence (present in <10% of participants) and with low relative abundance (mean relative abundance <0.01%) from unrarefied microbiota samples. This filtering was performed separately for the 3-month and 12-month microbiota samples, and it resulted in a decrease from 783 ASVs to 85 ASVs and from 783 ASVs to 105 ASVs at 3 months and at 12 months, respectively.

We assessed ASV differential abundance using two statistical approaches to identify consensus agreement. We first used the Analysis of Compositions of Microbiomes with Bias Correction (ANCOM-BC) [36]. This is a log-linear compositional modeling approach that attempts to correct for sampling fraction bias. We used default parameters but set the p-adjustment method to false discovery rate (FDR) and the prevalence cut-off to 0. For our second approach, we used the logistic compositional analysis (LOCOM) method [37]. This is a compositional multivariate logistic regression modeling approach that attempts to minimize experimental biases. We used default parameters but set the prevalence cut-off to 0 and the Firth threshold to 1. For our primary approach (ANCOM-BC), we considered significant taxa as those which reached the FDR q value threshold < 0.05; for our secondary approach (LOCOM), we considered significant taxa as those which reached the FDR q value threshold < 0.20.

### 2.11. SCFA Analyses

We used multiple linear regression models to examine the association of plastic bottle frequency at 3 months (<every feeding vs. every feeding) with SCFA concentrations as dependent variables at 3 (*n* = 65) and 12 months (*n* = 46). Models were run separately for each SCFA and for total SCFAs. We then examined the association of 3-month plastic bottle frequency (<every feeding vs. at every feeding) with each SCFA and total SCFAs across the 3- and 12-month timepoints using linear mixed models (*n* = 66) and Model 3 adjustments. We included terms for a random intercept and a time-by-plastic bottle use interaction and estimated the fixed effects using the restricted maximum likelihood approach.

### 2.12. Sensitivity Analyses

To determine the robustness of our findings to confounding, we repeated our above multiple linear regression analyses restricted to infants whose mothers (1) were not smokers at 3 months, (2) did not consume antibiotics during pregnancy, (3) had a post-secondary educational attainment, and (4) among infants who were exposed to any breast milk at 3 months of age. These analyses were conducted in the fully adjusted model (Model 3).

To assess the impact of infant diet—a key confounder—on outcomes, we restricted our analysis to infants fed with plastic bottles at every feeding at 3 months and compared the association of exclusive formula feeding vs. mixed feeding using linear regression models. These models were adjusted for Model 2 covariates and the duration of exclusive breast milk feeding.

All statistical analyses were performed using R statistical software (version 4.2.1). Unless otherwise specified, all statistical significance was based on a *p*-value of <0.05.

## 3. Results

### 3.1. Plastic Bottle Frequency and Infant Anthropometry

We present participant characteristics in Table 1. Of the 442 infants included, 284 (64.3%) were identified by mothers as Black and 256 (57.9%) were from households that made <$20,000/year. At 3 months, 299 (67.6%) of infants were plastic bottle fed at every feeding, and 226 (51.1%) infants were still being fed some breast milk.

We report associations of plastic bottle frequency at 3 months with anthropometric outcomes measured at 12 months in Appendix A. Infants who were plastic bottle fed 1–3 times/day at 3 months had a lower length-for-age z-score at 12 months before and after adjustment for key covariates (adjusted β = −0.45, 95% CI: −0.76, −0.13) compared to infants who used a plastic bottle every day; however, this association was attenuated and no longer statistically significant after adjustment for infant diet. Plastic bottle frequency was not associated with any other anthropometric outcome at 12 months, and these results were consistent in subgroup sensitivity analyses (Appendix A). Among infants who were plastic bottle fed at every feeding (Appendix A), triceps skinfolds were lower among infants with mixed feeding (formula and breast milk) compared to infants with exclusive formula feeding at 3 months (β = −0.61, 95% CI: −1.23, 0.00).

Given a lack of dose–response relationship for plastic bottle frequency and anthropometric outcomes, we assessed longitudinal growth trajectories by 3-month plastic bottle frequency categorized as <every feeding vs. every feeding. We present findings by timepoint in Appendix A, and trajectories based on linear mixed models are plotted in Figure 1. Only length-for-age z-score trajectories significantly differed by plastic bottle frequency (interaction *p* < 0.0001), such that infants who used plastic bottles less than every feeding showed a decrease in length-for-age z-score while infants who used plastic bottles at every feeding maintained a similar length-for-age z-score across timepoints.

### 3.2. Plastic Bottle Frequency and Gut Microbiota Diversity

Infants who used plastic bottles less frequently at 3 months had lower alpha diversity metrics at 3 months of age compared to infants who used plastic bottles every feeding; however, this association was attenuated and non-significant after adjustment for infant diet (Appendix A). There were no significant associations between plastic bottle frequency at 3 months and microbiome alpha diversity at 12 months (Appendix A). Results were similar in subgroup sensitivity analyses (Appendix A) and when comparing by feeding method among infants who used plastic bottles at every feeding (Appendix A). In linear mixed models, we report significant interactions for Pielou evenness and Shannon index such that infants fed plastic bottles at less than every feeding started with lower alpha diversity metrics at 3 months but increased to similar levels of alpha diversity metrics at 12 months when compared to infants fed with plastic bottles at every feeding (Figure 2).

Plastic bottle frequency at 3 months was associated with all beta diversity metrics at 3 months in unadjusted PERMANOVA models; however, these results were not significant after adjustment for key covariates and infant diet (Appendix A, Appendix A). Plastic bottle frequency at 3 months was not significantly associated with beta diversity metrics at 12 months.

### 3.3. Plastic Bottle Frequency and Gut Microbiota Differential Abundance

We report fully adjusted (Model 3) effect sizes for ASVs (at 3 months or 12 months) that were differentially abundant by 3-month plastic bottle frequency according to ANCOM-BC or LOCOM (Table 2). ANCOM-BC and LOCOM did not jointly identify differentially abundant ASVs at 3 or 12 months. In the 3-month microbiota, LOCOM identified significantly lower *Collinsella aerofaciens* among infants who used plastic bottles less than every feeding vs. at every feeding. In the 12-month microbiota, ANCOM-BC identified significantly lower *Parabacteroides merdae* and *Faecalibacterium (ASV110)* among infants who used plastic bottles less than every feeding at 3 months.

### 3.4. Plastic Bottle Frequency and Short-Chain Fatty Acids

Infants fed with a plastic bottle less than every feeding at 3 months had a significantly lower isovaleric acid concentration at 3 months of age; however, this association was attenuated and became non-significant after adjustment for infant diet (Table 3). Plastic bottle frequency at 3 months was not associated with any other SCFA nor with total SCFAs at either timepoint (Table 3). Results were consistent in sensitivity analyses (Appendix A). Among infants with 3-month plastic bottle use at every feeding, having any breast milk vs. no breast milk at 3 months was associated with lower valeric acid at 3 months and with higher butyric acid at 12 months (Appendix A); however, we caution interpretation of the 12-month findings due to the small sample sizes. In linear mixed models, there were no significant interactions between 3-month plastic bottle frequency and time for any SCFAs (Figure 3).

## 4. Discussion

In this birth cohort of racially diverse mother–infant dyads from North Carolina, plastic bottle frequency at 3 months was not strongly associated with measures of growth or adiposity in the first year of life. In addition, less frequent plastic bottle use at 3 months was associated with lower 3-month fecal microbiota diversity and of isovaleric acid levels; however, these associations were partially attenuated by adjustment for infant diet.

To our knowledge, no other studies have examined the associations or effects of plastic bottle use on measures of infant growth or adiposity, but studies have reported on associations of plastic chemicals. A prospective study in children from New York found that urinary phthalates, a group of chemicals used to make plastics more durable, were associated with greater BMI and waist circumference over an average of 1.1 years of follow-up [6]. However, compared to our study, the New York study measured urinary phthalates rather than plastic bottle use, examined older children (aged 6 to 8 years at baseline), and had a demographically distinct population (e.g., greater prevalence of Hispanic ethnicity), which may have contributed to the divergent findings. Additionally, in a cross-sectional study among Chinese school children, overweight or obese children had higher levels of urinary phthalates compared to children who were normal weight [38]. In contrast to our largely null results in infants, the positive results from these pediatric studies could indicate that the sensitivity period may be in later life or that there is a stronger effect of plastic chemicals on anthropometric growth. Dietary phthalates have also been shown to increase weight, lipids, triglycerides, and disrupt microbial morphology and composition in murine models [39], however, the relative duration of phthalate exposure (14 weeks) and dosage (5 mg/kg body weight) in mice was greater than the plastic exposure assessed in our study.

We are also not aware of other studies in human infants that have examined the association of plastic bottle usage with fecal microbiota features. In mice, introducing polystyrene microplastics altered gut microbiota alpha diversity and composition [40,41] and altered microbial metabolites (increased bile acids; decreased acetic, propionic, butyric, and isobutyric acids) [41]. Similarly, adult [42,43] and infant [44] artificial intestinal models suggest that realistic microplastic exposures could impact microbiota composition and function. While not in infants, two small cross-sectional studies in adults have examined plastic exposure with the gut microbiome: one among Indonesian adults (N = 22) [45] and the other comparing Chinese plastics factory workers to non-plastics factory workers (N = 20 per group) [46]. Neither study identified associations of microplastics with microbial alpha diversity metrics, but they did identify correlations of plastic exposure with specific taxa. In our study, we did not find significant differences in infant microbiota diversity or in SCFAs by plastic bottle frequency, after we controlled for infant diet. Although our study was limited by sample size, it represents the largest analysis of a plastic-product exposure with fecal microbiota and microbial metabolites, the only longitudinal analysis, and the only analysis among infants.

Our study is not without limitations. First, despite our prospective collection of data, our measure of plastic bottle frequency was based on self-report by mothers and could be subject to reporting bias. Additionally, we did not collect information on the type of plastic materials used in the bottle, nor do we have more objective measures of plastic exposure (e.g., biomarkers). We are also unable to compare plastic vs. glass bottle use, which would be an interesting comparison for future studies. Secondly, although we adjusted for infant feeding method and duration of exclusive breastmilk feeding, and we conducted subgroup sensitivity analyses, we cannot rule out influences from other dietary aspects (e.g., formula type) with our study design and sample size. Third, we did not have blood measurements available to measure circulating SCFAs, which could offer different insights compared to stool SCFAs. Fourth, we relied on 16S rRNA sequencing data, which may not always allow for species-level resolution. Lastly, despite covariate adjustment, we cannot rule out the possibility of unmeasured or residual confounding.

It is important to study the effects of plastic exposure in early life, as this is a critical window for child growth and development. Plastics comprise thousands of chemicals, some of which (e.g., bisphenol A, phthalates) have endocrine-disrupting effects [47]. Plastics can also degrade into absorbable microplastics and nanoplastics, which may promote inflammation in addition to transporting plastic chemicals [48]. Thus, plastic exposure in early life could adversely impact growth and health in developing infants. Furthermore, it is important to study plastic exposure in early life as infants and children may have unique concentrations of and exposures to plastics. A recent study found infants had higher levels of fecal microplastics compared to adults, suggesting that they have greater dietary plastic exposures [2]. Compared to adults, differences in behavioral factors (e.g., mouthing, eating frequency, hand-to-mouth transfers) may result in differences in plastic exposure among infants [48]. Additionally, plastic bottles are a unique source of plastics exposure in infancy which may drive the greater dietary plastic exposure. Li et al. estimated that infants on average are exposed to over 1 million microplastic particles per capita per day from plastic bottle use [3]. Given the widespread use of plastic bottles and potential adverse health effects of plastics exposures in development, it is important to understand the potential implications of plastic exposure during the early life period.

In conclusion, we did not find associations between plastic bottle frequency at 3 months and infant growth and adiposity over the first year of life, and associations of plastic bottle exposure with fecal microbiota and SCFAs over the first year of life were largely attenuated by measures of infant diet. Larger longitudinal birth cohort studies, with more robust analysis of the concentrations and types of plastic particles, and with greater detailed infant dietary information, are required to investigate whether early life plastic exposure can influence infant gut microbiota and health outcomes.

## Figures and Tables

**Figure 1 microorganisms-11-02924-f001:**
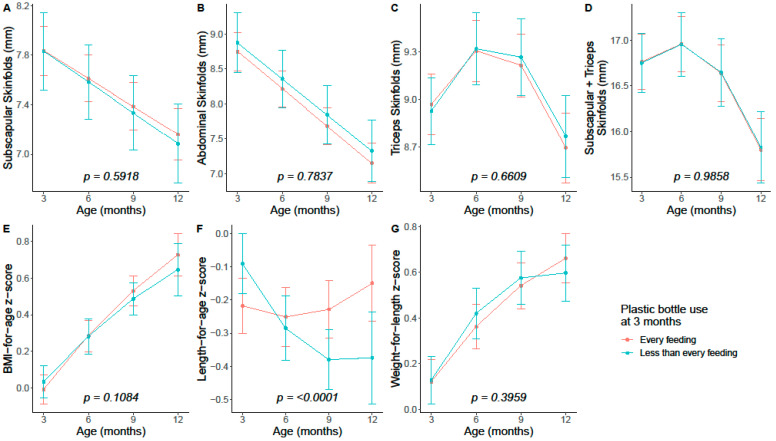
Anthropometric growth trajectories by 3-month plastic bottle frequency among infants *n* = 442 from the Nurture birth cohort. Estimates (95% confidence intervals) were based on multivariable linear mixed effect models. Multivariable models included variables from Model 3 and an interaction term for time-by-plastic bottle feeding frequency at 3 months. Quadratic fits were considered if it significantly improved model fit (likelihood ratio test *p* < 0.05). Reported interaction *p*-values are for the differences in linear trend. (**A**) Subscapular skinfolds; (**B**) Abdominal skinfolds; (**C**) Triceps skinfolds; (**D**) Sum of subscapular and triceps skinfolds; (**E**) BMI-for-age z-score; (**F**) Length-for-age z-score; (**G**) Weight-for-length z-score.

**Figure 2 microorganisms-11-02924-f002:**
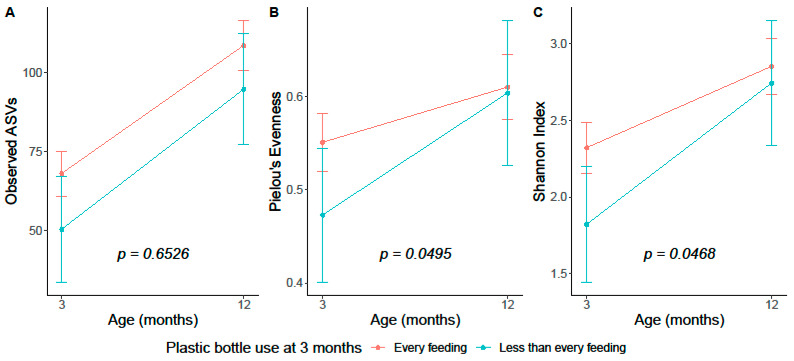
Fecal microbiota alpha diversity metrics at 3 months and at 12 months of age by 3-month plastic bottle frequency among *n* = 63 infants from the Nurture birth cohort. Estimates (95% confidence intervals) were based on multivariable linear mixed effect models with fecal microbiota alpha diversity metrics measured at 3 months and 12 months. Multivariable models included variables from Model 3 and an interaction term for time-by-plastic bottle feeding frequency at 3 months. Interaction term *p*-values are reported. (**A**) Observed ASVs. This model additionally adjusted for sequence depth. (**B**) Pielou’s evenness. (**C**) Shannon index. Abbreviations: ASV = amplicon sequencing variant.

**Figure 3 microorganisms-11-02924-f003:**
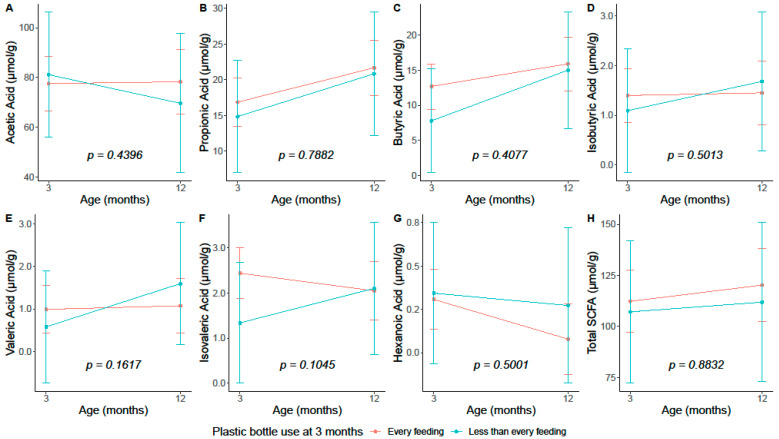
Fecal SCFAs at 3 months and at 12 months of age by 3-month plastic bottle use frequency among *n* = 66 infants from the Nurture birth cohort. Estimates (95% confidence intervals) were based on multivariable linear mixed effect models with fecal SCFAs measured at 3 months and 12 months as the dependent variables. Multivariable models included variables from Model 3 and an interaction term for time-by-plastic bottle feeding frequency at 3 months. Interaction term p-values are reported. (**A**) Acetic acid. (**B**) Propionic acid. (**C**) Butyric acid. (**D**) Isobutyric acid. (**E**) Valeric acid. (**F**) Isovaleric acid. (**G**) Hexanoic acid. (**H**) Total fecal SCFAs. Abbreviations: CI = confidence interval, SCFA = short-chain fatty acid.

**Table 1 microorganisms-11-02924-t001:** Characteristics of mothers and infants from the Nurture birth cohort at 3 months after birth, overall and stratified by plastic bottle feeding frequency at 3 months of age.

	Overall (N = 442)	Every Feeding (N = 299)	4+/Day (N = 42)	1–3/Day (N = 59)	<1/Day (N = 42)
Delivery mode (N(%))					
Cesarean delivery	155 (35.1%)	113 (37.8%)	16 (38.1%)	20 (33.9%)	6 (14.3%)
Operative vaginal delivery (vacuum or forceps)	11 (2.5%)	6 (2.0%)	1 (2.4%)	3 (5.1%)	1 (2.4%)
Spontaneous vaginal delivery	275 (62.2%)	179 (59.9%)	25 (59.5%)	36 (61.0%)	35 (83.3%)
Missing	1 (0.2%)	1 (0.3%)	0 (0%)	0 (0%)	0 (0%)
Pre-pregnancy BMI (kg/m^2^)	30.4 (9.5)	31.4 (10.0)	31.2 (9.6)	27.6 (6.3)	26.1 (8.5)
Missing	2 (0.5%)	2 (0.7%)	0 (0%)	0 (0%)	0 (0%)
Mother age (years)	27.6 (5.7)	26.8 (5.4)	29.0 (5.6)	30.0 (6.5)	29.3 (5.6)
Baby sex, Female (N(%))	221 (50.0%)	150 (50.2%)	21 (50.0%)	31 (52.5%)	19 (45.2%)
Baby race, Black (N(%))	284 (64.3%)	216 (72.2%)	25 (59.5%)	32 (54.2%)	11 (26.2%)
Missing	6 (1.4%)	5 (1.7%)	1 (2.4%)	0 (0%)	0 (0%)
Baby feeding status at 3 months (N(%))					
Mixed breast milk and formula fed	158 (35.7%)	89 (29.8%)	28 (66.7%)	31 (52.5%)	10 (23.8%)
Exclusive breast milk fed	68 (15.4%)	3 (1.0%)	7 (16.7%)	26 (44.1%)	32 (76.2%)
Exclusive formula fed	216 (48.9%)	207 (69.2%)	7 (16.7%)	2 (3.4%)	0 (0%)
Total months of any (exclusive or mixed) breast milk feeding (months)	3.00 [1.00, 3.00]	2.00 [0, 3.00]	3.00 [3.00, 3.00]	3.00 [3.00, 3.00]	3.00 [3.00, 3.00]
Total months of exclusive breast milk feeding (months)	0 [0, 1.00]	0 [0, 0]	0 [0, 1.75]	1.00 [0, 3.00]	2.50 [2.00, 3.00]
Total months of exclusive formula feeding (months)	0 [0, 2.00]	1.00 [0, 3.00]	0 [0, 0]	0 [0, 0]	0 [0, 0]
Mother educational attainment (N(%))					
High school or less	197 (44.6%)	161 (53.8%)	17 (40.5%)	11 (18.6%)	8 (19.0%)
More than high school	245 (55.4%)	138 (46.2%)	25 (59.5%)	48 (81.4%)	34 (81.0%)
Household income (N(%))					
<$20k	256 (57.9%)	199 (66.6%)	21 (50.0%)	26 (44.1%)	10 (23.8%)
>=$20k	186 (42.1%)	100 (33.4%)	21 (50.0%)	33 (55.9%)	32 (76.2%)
Mother was on antibiotics during pregnancy (N(%))	139 (31.4%)	95 (31.8%)	15 (35.7%)	11 (18.6%)	18 (42.9%)
Mother currently smokes (N(%))	90 (20.4%)	81 (27.1%)	4 (9.5%)	3 (5.1%)	2 (4.8%)
Missing	66 (14.9%)	52 (17.4%)	5 (11.9%)	6 (10.2%)	3 (7.1%)
Baby gestational age (weeks)	38.7 (1.5)	38.6 (1.5)	38.8 (1.8)	38.9 (1.4)	38.7 (1.4)
Baby birth weight (kg)	3.2 (0.5)	3.2 (0.5)	3.2 (0.4)	3.4 (0.5)	3.3 (0.5)
Number of persons in household (N)	3 [2, 4]	3 [2, 5]	3 [2, 4]	3 [2, 4]	3 [2, 4]
Missing	5 (1.1%)	5 (1.7%)	0 (0%)	0 (0%)	0 (0%)

Descriptive statistics are provided for *n* = 442 of the *n* = 666 infant–mother pairs from the Nurture birth cohort who had complete data on plastic bottle frequency at 3 months of age, birth weight (kg), gestational age (weeks), maternal age (years), household income (<$20,000 per year vs. ≥$20,000 per year), duration of exclusive breast milk exposure (months), and feeding status at 3 months (exclusive breast milk, exclusive formula, or mixed feeding). Characteristics are presented as mean (standard deviation), median (interquartile range), or N (%).

**Table 2 microorganisms-11-02924-t002:** Significantly differentially abundant fecal microbiota amplicon sequence variants (ASVs) and effect sizes for infants who are plastic bottle fed less than every feeding at 3 months (compared to infants who are plastic bottle fed at every feeding).

Timepoint	Taxa Name	ASV Prevalence	ASV Relative Abundance	ANCOM-BC			LOCOM		
				*p*-Value	FDR *q* Value	Effect Size ^†^	*p*-Value	FDR q Value	Effect Size ^‡^
3	*Collinsella aerofaciens* (ASV6)	69.23%	4.63%	0.0219	0.4653	−2.51	0.0020	**0.1700**	−7.50
12	*Parabacteroides merdae* (ASV18)	48.98%	0.72%	<0.0001	**0.0002**	−3.13	0.7530	0.8990	−2.57
12	*Faecalibacterium* (ASV110)	16.33%	0.18%	0.0007	**0.0383**	−2.07	0.1860	0.7770	−14.30

A total of *n* = 62 infants (at month 3) and *n* = 45 infants (at month 12) were included in the analyses. The ASV prevalence and relative abundance within the sample are provided. Significantly differentially abundant ASVs were identified using the Analysis of Compositions of Microbiomes with Bias Correction (ANCOM-BC) and logistic compositional analysis (LOCOM) differential abundance models; these multivariable models included adjustment for variables in Model 3. **Bold text** indicates statistically significant differentially abundant ASVs for each method, i.e., if they met an FDR threshold of *q* < 0.05 for ANCOM-BC or an FDR threshold *q* < 0.20 for LOCOM. ^†^: The ANCOM-BC effect size is interpreted as the log-fold difference in absolute ASV abundance; a negative value indicates a lower abundance among infants who are plastic bottle fed less than every feeding. ^‡^: The LOCOM effect size is interpreted as the log odds that a microbiota sequencing read falls into the taxon of interest; a negative value indicates a lower abundance among infants who are plastic bottle fed less than every feeding. Abbreviations: ANCOM-BC = Analysis of Compositions of Microbiomes with Bias Correction, ASV = amplicon sequencing variant, FDR = false discovery rate, LOCOM = logistic compositional analysis.

**Table 3 microorganisms-11-02924-t003:** Unadjusted and multivariable-adjusted mean (95% confidence intervals) differences in fecal short-chain fatty acids (SCFAs) according to plastic bottle frequency at 3 months of age, among infants from the Nurture birth cohort, at 3 months and at 12 months of age.

Outcome		3 Months		12 Months	
		Plastic Bottle Provided Every Feeding	Plastic Bottle Provided Less than Every Feeding	Plastic Bottle Provided Every Feeding	Plastic Bottle Provided Less Than Every Feeding
		N = 51	N = 14	N = 35	N = 11
Total SCFAs (μmol/g)	M1	-	−12.42 (−40.93, 16.08)	-	−19.53 (−54.00, 14.94)
	M2	-	−13.33 (−53.68, 27.02)	-	−25.08 (−71.04, 20.88)
	M3	-	3.73 (−45.31, 52.77)	-	−33.43 (−146.34, 79.48)
Acetic Acid (μmol/g)	M1	-	−1.26 (−22.09, 19.56)	-	−15.07 (−38.91, 8.76)
	M2	-	3.87 (−25.71, 33.45)	-	−19.82 (−53.34, 13.71)
	M3	-	12.65 (−23.10, 48.39)	-	−24.74 (−101.78, 52.3)
Propionic Acid (μmol/g)	M1	-	−2.56 (−8.29, 3.18)	-	−1.11 (−9.89, 7.68)
	M2	-	−7.85 (−15.84, 0.13)	-	−1.60 (−12.80, 9.60)
	M3	-	−4.91 (−14.34, 4.52)	-	0.51 (−15.24, 16.26)
Butyric Acid (μmol/g)	M1	-	**−6.30 (−11.87, −0.73)**	-	−3.16 (−11.11, 4.79)
	M2	-	−6.02 (−13.73, 1.69)	-	−2.95 (−13.68, 7.77)
	M3	-	−2.28 (−11.77, 7.21)	-	−10.48 (−25.56, 4.60)
Isobutyric Acid (μmol/g)	M1	-	0.11 (−0.96, 1.19)	-	0.55 (−0.53, 1.64)
	M2	-	−0.59 (−2.11, 0.93)	-	0.79 (−0.57, 2.15)
	M3	-	−0.21 (−2.11, 1.68)	-	0.01 (−2.00, 2.02)
Valeric Acid (μmol/g)	M1	-	−0.85 (−1.94, 0.24)	-	−0.18 (−1.44, 1.08)
	M2	-	−0.69 (−1.82, 0.44)	-	−0.67 (−2.13, 0.80)
	M3	-	−0.48 (−2.34, 1.39)	-	0.79 (−1.28, 2.86)
Isovaleric Acid (μmol/g)	M1	-	**−1.58 (−2.65, −0.52)**	-	−0.70 (−2.02, 0.63)
	M2	-	**−2.12 (−3.64, −0.60)**	-	−1.00 (−2.57, 0.57)
	M3	-	−1.10 (−2.95, 0.74)	-	0.37 (−1.82, 2.56)
Hexanoic Acid (μmol/g)	M1	-	0.01 (−0.40, 0.42)	-	0.15 (−0.03, 0.32)
	M2	-	0.07 (−0.49, 0.63)	-	0.17 (−0.07, 0.40)
	M3	-	0.07 (−0.64, 0.77)	-	0.11 (−0.24, 0.45)

A total of *n* = 65 infants (at 3 months of age) and *n* = 46 infants (at 12 months of age) were included in the analyses. Estimates (95% CI) were based on multivariable linear regression models with individual and total SCFAs as the dependent variables; models were analyzed separately by SCFA and by timepoint. **Bold text** indicates statistical significance at *p* < 0.05. Models were analyzed according to the following schema: M1: unadjusted; M2: M1 + birth weight (kg), gestational age (weeks), maternal age (years), household income (<$20,000 per year vs. ≥$20,000 per year); M3: M2 + duration of exclusive breast milk exposure (months), current feeding status at 3 months (exclusive breast milk, exclusive formula, or mixed feeding); Abbreviations: CI = confidence interval, SCFA = short-chain fatty acid.

## Data Availability

The data that support the findings of this study are available from the Nurture birth cohort principal investigator (PI Benjamin-Neelon). Some restrictions apply to the availability of these data, which were used under license for the current study. Data may be made available from the authors upon reasonable request and with permission from the Duke University Medical Center.

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
