# Peer review of "Associations of Plastic Bottle Exposure with Infant Growth, Fecal Microbiota, and Short-Chain Fatty Acids"

_microorganisms, 2023, doi:10.3390/microorganisms11122924_

Round 1
Reviewer 1 Report
Comments and Suggestions for Authors
The authors tried to prove the association of using plastic bottles with the growth, fecal microbiota, and SCFAs. The manuscript is generally well written, in a good quality English language. Some changes must be made to improve its quality.
The title is somehow misleading as the first conclusion is that there is no association with infant growth.
The authors should verify the use of the abbreviations (BPA in the abstract, BMI in lin e45, for example)
In the Methods, the small paragraph on Statistical analysis may be moved at the end of the methods or left out as there are referrals about analysis in all the other paragraphs.
In the results presentations, too many important data are presented only in Supplementary Tables. Moreover, the Tables and Figures have too long legends. Including the text from the legend in the text would be better. Some are already presented in Methods (see Models). The legends are extremely long for fig 1, fig 2, and tables 2 and 3.
The discussion sections seem too short, and besides the limitations, the authors should also discuss the importance of their study and the results obtained. This section should be improved.
Comments on the Quality of English Language
The quality of the English language is very good; just some minor editing issues may need to be verified.
Author Response
[Reviewer 1] Comment 1: The title is somehow misleading as the first conclusion is that there is no association with infant growth.
Response: We thank the reviewer for this suggestion. To reduce confusion, we have updated title to be: “Evaluating Associations of Plastic Bottle Exposure with Infant Growth, Fecal Microbiota, and Short-Chain Fatty Acids”
[Reviewer 1] Comment 2: The authors should verify the use of the abbreviations (BPA in the abstract, BMI in lin e45, for example)
Response: We thank the reviewer for this comment and have included the definitions of all abbreviations.
[Reviewer 1] Comment 3: In the Methods, the small paragraph on Statistical analysis may be moved at the end of the methods or left out as there are referrals about analysis in all the other paragraphs.
Response: We have now moved this section to the end of the statistical analysis.
[Reviewer 1] Comment 4: In the results presentations, too many important data are presented only in Supplementary Tables. Moreover, the Tables and Figures have too long legends. Including the text from the legend in the text would be better. Some are already presented in Methods (see Models). The legends are extremely long for fig 1, fig 2, and tables 2 and 3.
Response: We thank you for the comment and we have shortened the legends in Figure 1,2, and Tables 2,3 by placing information about model covariates in the methods of the paper. We also agree that there is important data from the supplementary, specifically on SCFA, which we have added to our main paper as Figure 3.
[Reviewer 1] Comment 5: The discussion sections seem too short, and besides the limitations, the authors should also discuss the importance of their study and the results obtained. This section should be improved.
Response: We thank the reviewer for this comment. We have enhanced our discussion by adding discussion on the hypothesized importance of plastics exposure for infant growth [lines 481-496], as well as some recent studies on plastic chemical exposure with obesity (in children and in mice) [lines 443-451].
Reviewer 2 Report
Comments and Suggestions for Authors
Thank you for submitting the manuscript "Associations of Plastic Bottle Exposure with Infant Growth, Fecal Microbiota, and Short-Chain Fatty Acids" to Microorganisms. The manuscript is interesting and the research appears to have been conducted with scientific rigor. The work describes the results obtained from a survey of infants in the first year of life and the relationship between anthropometric parameters, fecal microbiota profile and the use of milk bottles.
- Consider revising the manuscript according to the journal's author instructions.
- Consider including a flowchart of how the work was carried out to make the study protocol clearer.
- Consider correcting any typos throughout the text.
- Consider improving the resolutions of the figures. As it is, it is practically impossible to read.
- Consider summarizing the title of the figures so that it can be explanatory, but shorter. Even in some tables, the title is long as in Table 3, abbreviation information is lost and it is not possible to critically analyze the results.
- A major issue of the work is that although the relationship between the presence of plastic in infant food and the microbiota and anthropometric parameters is evaluated, no phthalates were evaluated in any human excreta, which leaves the question open as to whether this relationship actually occurs.
- For this reason, the discussion is still very superficial and is the weak point of the work.
Comments on the Quality of English LanguageMinor editing of English language required.
Author Response
[Reviewer 2] Comment 6: Consider revising the manuscript according to the journal's author instructions.
Response: We have updated our manuscript to align with the journal submission guidelines.
[Reviewer 2] Comment 7: Consider including a flowchart of how the work was carried out to make the study protocol clearer.
Response: We have included a study flowchart from the supplementary materials (Supplementary Figure S1). We are happy to include this as a main figure per the editor’s discretion.
[Reviewer 2] Comment 8: Consider correcting any typos throughout the text.
Response: We have reviewed the manuscript to address potential typos.
[Reviewer 2] Comment 9: Consider improving the resolutions of the figures. As it is, it is practically impossible to read.
Response: We apologize that the figure resolutions did not come through. We have reformatted figures to a greater resolution.
[Reviewer 2] Comment 10: Consider summarizing the title of the figures so that it can be explanatory, but shorter. Even in some tables, the title is long as in Table 3, abbreviation information is lost and it is not possible to critically analyze the results.
Response: We thank the reviewer for this comment and have shortened titles and legends for figures/tables (see Comment 4)
[Reviewer 2] Comment 11: A major issue of the work is that although the relationship between the presence of plastic in infant food and the microbiota and anthropometric parameters is evaluated, no phthalates were evaluated in any human excreta, which leaves the question open as to whether this relationship actually occurs. For this reason, the discussion is still very superficial and is the weak point of the work.
Response: We agree with the reviewer that the lack of objective measures of plastic exposure is a limitation of our study, and we have included this in our limitations section (lines 470-472). However, given the dearth of data on this topic and the increasing importance of understanding the role of plastic exposure in the health of developing infants, we believe our longitudinal work is an important step in exploring these associations. We have further emphasized phthalates in the expansion of our discussion section (see Comment 5). We have augmented discussion to describe our results within the context of what is known about phthalates and adiposity, microbiome, and metabolome.
Reviewer 3 Report
Comments and Suggestions for Authors
The manuscript is very interesting and outlines an important issues concerning fecal microbiota in children from 3 months to 12 months. it presents important results concerning sequencing the microbiota. Some points that the authors should consider is if the patients needed antibiotics in the 12 month follow up, also if mothers administrated pro-biotics to children during the study period and if they used the same type of formula. For example if some kids needed hipoalergenic formulas, that could represent a bias in comparing the lots.
All in all the study is very interesting and covers a new and interesting issue that should be furthered studied. Congratulations for the ideea and for the design. It should be published due to the scarce literature concerning differentiations on microbiota in children, and moreover the genetic study on the microbiota that presents statistical significant correlations.
congrats and good luck.
Comments on the Quality of English LanguageThe quality of English language needs minor revision.
Author Response
[Reviewer 3] Comment 12: Some points that the authors should consider is if the patients needed antibiotics in the 12 month follow up, also if mothers administrated pro-biotics to children during the study period and if they used the same type of formula. For example if some kids needed hipoalergenic formulas, that could represent a bias in comparing the lots.
Response: We thank the reviewer for raising these considerations. We do not have information on probiotic supplementation or formula type (e.g., hypoallergenic) in this population. We did not control for infant antibiotic use at 12 months, as we did not view it as an exposure that was causally associated with plastic bottle use (and it occurred after plastic bottle use).